

# The stability of soil organic carbon across topographies in a tropical rainforest

Yamin Jiang[1,*], Huai Yang[2,3,*], Qiu Yang[1], Wenjie Liu[1,4], Zhaolei Li[4], Wei Mao[1], Xu Wang[1] and Zhenghong Tan[1]

[1] Hainan University, College of Ecology and Environment, Haikou, China
[2] International Center for Bamboo and Rattan, BeiJing, China
[3] Chinese Academy of Forestry, Jianfengling National Key Field Research Station for Tropical Forest Ecosystem, Research Institute of Tropical Forestry, Hainan, China
[4] Northern Arizona University, Center for Ecosystem Science and Society, Flagstaff, AZ, USA
[*] These authors contributed equally to this work.

## ABSTRACT

Mechanisms of soil organic carbon (SOC) stability are still unclear in forest ecosystems. In order to unveil the influences of topography on the SOC stability, a 60ha dynamic plot of a tropical montane rainforest was selected in Jianfengling, in Hainan Island, China and soil was sampled from 60 quadrats. The chemical fractions of the SOC were detected with 13C CPMAS/NMR and path analyses explore the mechanisms of SOC stability in different topographies. The chemical fractions of the SOC comprised alkyl carbon > O-alkyl carbon > carboxyl carbon > aromatic carbon. The decomposition index (DI) values were greater than 1 in the different topographies, with an average DI value was 1.29, which indicated that the SOC in the study area was stable. Flat and top areas (together named RF) had more favorable nutrients and silt contents compared with steep and slight steep areas (together named RS). The influencing factors of SOC stability varied across the topographies, where SOC, soil moisture (SM) and ammoniacal nitrogen ($NH_4^+$-N, AN) were the main influencing factors in the RF, while SM and AN were the main factors in the RS. Greater SOC and AN strengthened the SOC stability, while higher soil moisture lowered SOC stability. The inertia index was higher in the RS than the RF areas, indicating that local topography significantly affects SOC content and SOC stability by changing soil environmental factors. Topography cannot be neglected in considering SOC stability and future C budgets.

## INTRODUCTION

The soil organic carbon (SOC) of forests plays an important role in the global carbon cycle (*Pan et al., 2011*). Changes in SOC storage can significantly impact the global carbon cycle and climate change (*Stockmann et al., 2013*). SOC stability is the ability of soil organic matter to resist disturbance and maintain its original carbon levels under the present conditions (*Wiesmeier et al., 2014*). High SOC stability would be beneficial for the accumulation of SOC (*Schmidt et al., 2011*). The SOC stabilisation mechanisms that are commonly accepted include chemical stabilisation, physical stabilisation, and biochemical

Corresponding authors
Wenjie Liu, liuwj@hainanu.edu.cn
Zhaolei Li, zhaoleilee@sdau.edu.cn

stabilisation (*Christensen, 1996*; *Fontaine et al., 2007*; *Six et al., 2002*). In terms of chemical stabilisation, SOC consists of chemical fractions of different stability and this has long been considered as the principal mechanism of SOC stability (*Yang et al., 2020*). The relationship between soil structure and the ability of soil to stabilize soil organic matter indicating that physical protection is a key element in SOC stability (*Six et al., 2002*). Studies on the protection of soil C by soil texture, such as silt and clay, had long been well established (*Hassink, 1997*). Solid 13CNMR technology is a method for dissecting SOC chemical fractions and has been applied into tropical forest soil stability research (*Chen, 2012*; *Shang et al., 2013*; *Wang, 2010*). Previous studies illustrated that the composition of the SOC chemical fractions is affected by soil microorganisms, soil properties, and carbon input (*Elberling, Breuning-Madsen & Knicker, 2013*; *González-Pérez et al., 2008*; *Ktigel-Knabner, 1997*; *Lorenz, Lal & Jiménez, 2010*). Topographic factors, such as elevation, slope, concavity and convexity, affect water flow and erosion, plants growth and litter decomposition, and hence influence SOC content and quality (*Fernández-Romero, Lozano-García & Parras-Alcántara, 2014*; *Sun, Zhu & Guo, 2015*). As such, there is inherent heterogeneity in SOC stability associated with the spatial variability of topographic factors. However, studies on factors influencing SOC stability have mainly focused on tree species, soil minerals protection or altitude (*Angst et al., 2018*) rather than topography.

Topography itself is a comprehensive factor, and it should be taken into consideration in SOC stability. Topography governs the allocation of water and heat resources, which affects the spatial allocation of vegetation and may thus affect the quantity and quality of SOC (*Sun, Zhu & Guo, 2015*). In addition, the movement of water and nutrients due to different topography types contributes to soil properties heterogeneity (*Tsui, Chen & Hsieh, 2004*). Studies found SOC fractions and stability varied across topography. Some studies found in lowland and plain areas alkyl carbon accounted for the most of the SOC fractions and indicated disturbance of soil and soil particle size be the key influencing factors (*Chen, Xu & Mathers, 2004*; *Shang et al., 2013*). However, other studies in plain and hills found a relative lower DI index (all less than 0.9) possibly due to the input of SOC and soil moisture (*Lorenz, Lal & Jiménez, 2010*; *Wang, 2010*). Studies in plateau found SOC that were not trapped in the iron nodules had a relative higher DI index (1.25) (*Elberling, Breuning-Madsen & Knicker, 2013*). We hypothesized that nutrients and soil environment factors would be the key influencing factors on SOC stability.

The Jianfengling montane rainforest is one of the best preserved tropical primaeval forests in China (*Li et al., 2012*). Previous studies have found that terrain heterogeneity in this area had a strong effect on the soil properties (*Shi, 2012*) and pointed out that SOC stability is significantly influenced by soil properties (*Liu et al., 2007*). Soil properties in different topographic positions also have distinctive characteristics under micro-climate conditions (*Zhu et al., 2014*), which fundamentally influences nutrients, hydrological processes, and thereby, the SOC stability across these topographies. However, there has been a systematic lack of studies specifically addressing how topographies influence the SOC stability by changing soil properties. The objectives of this study were to investigate the differences in soil properties and SOC stability at four topographic positions, and to identify the mechanisms of SOC stability in different topographies in tropical rainforests.
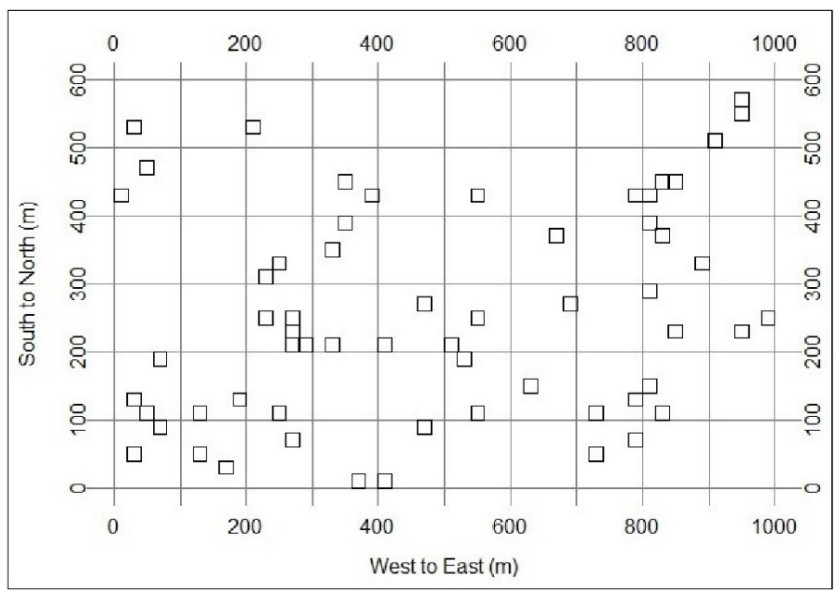

**Figure 1** Distribution of soil sampling sites in the 60 ha plot in tropical forests.

To address these objectives, we collected soils from 60 quadrats in a 60ha dynamic plot in different topographical areas in the Jiangenling tropical montane rainforest. The stability of the SOC was demonstrated by SOC chemical fractions analysed by 13C-NMR.

## MATERIALS AND METHODS

The study area was a 60 ha dynamics plot in the tropical montane rain forest (18.23°–18.50°N, 108.36°–109.05°E) with an elevation of 800–1,000 m, in Jianfengling, the southwest region of Hainan Island, China(*Li & Zhou, 2002*). The climate of the studied area is tropical and rainy climate, with a mean annual precipitation of 1,700–2,600 mm, and a mean annual temperature of 20–25 °C (*Yang et al., 2016*). The main plant families are Lauraceae, Rubiaceae, Fagaceae, Myrtaceae (*Li & Zhou, 2002*) and the soil belongs to the Ultisols order according to the USDA Soil Taxonomy.

Based on the technical specifications of the Centre for Tropical Forest Science, Smithsonian Institute, the 60 ha dynamics plot was divided into 1,500 quadrants, each 20 m × 20 m in size. Figure 1 shows the distribution of the 60 randomly selected soil sampling sites from the 1,500 quadrats. In each sampling site, the cutting ring method was used to obtain topsoil (0–10 cm) samples (each sample including four replicates). In total, 240 soil samples were collected and brought back to the laboratory. A part of each soil sample was stored at 4 °C to measure acid phosphatase activity (APA). A part was air-dried and sieved over 2 mm to characterize soil particle size (sand, silt and clay content). The rest of soil sample was air-dried and ground into 1 mm to analyse soil total nitrogen (TN), total phosphorus (TP), available phosphorus (AP), soil pH, and SOC chemical fractions.

Soil samples were pretreated with 10% (v/v) hydrofluoric acid (HF) solutions before solid-state 13C-CPMAS NMR analysis, to remove $Fe^{3+}$ and $Mn^{2+}$ in the soil, improving

the signal/noise ratio. In detail, 10 g ground soil sample was vibrated for 2 h with 50 ml HF solutions and then centrifuged (3,000 rpm) for 10 min to separate the suspension. The above steps were repeated five times to obtain the precipitate (*Mathers et al., 2002*). The precipitate was washed 5 times with 50 ml deionized water to remove residual HF, then freeze-dried for further analysis. The fractions of SOC were measured with a 13C CPMAS/NMR, 4 mm probe pulse sequence. The CPTOSS 13C resonant frequency was 100.38 MHz, the 1 H resonant frequency was 399.16 MHz, rotation speed was 5 kHz, the contact time was 3 ms and cycle time was 1 s with a sampling frequency of 4 k. According to a previous study (*Baldock et al., 1992*), different peaks represent different C compounds: 0–45 ppm is alkyl carbon, 45–110 ppm is O-alkyl carbon, 110–160 ppm is aromatic carbon, and 160–220 ppm is carbonyl carbon. A total of 240 soil samples were analysed in the Analytical Centre of the Institute of Chemistry Chinese Academy of Sciences (Beijing City). MestRe-C software was used to analyse the chemical fractions of the SOC. We used two indices, the decomposition index (DI) and the inertia index (II), to represent SOC stability.

$$DI = \frac{\text{alkyl carbon}}{\text{O} - \text{alkyl carbon}} \tag{1}$$

$$II = \frac{\text{alkyl carbon} + \text{aromatic carbon}}{\text{o} - \text{alkyl carbon} + \text{carboxyl carbon}} \tag{2}$$

The DI can reflect microbial processing with higher ratios indicating losses of more labile C relative to poorer-quality C compounds (*Cusack et al., 2011*), therefore, the higher ratio, the more stable is the SOC (*Chen, 2012*; *Ostertag et al., 2008*). Higher II values mean more alkyl carbon and aromatic carbon in the SOC, indicating higher SOC stability (*Ostertag et al., 2008*), which is beneficial for SOC accumulation.

Soil organic carbon and total N contents were determined by the $K_2Cr_2O_7$–$H_2SO_4$ oxidation method and the Kjeldahl procedure respectively (*Liu et al., 2012*). Soil total phosphorus (TP) was extracted by the semimicro kelvin method and measured by an automatic flow analyzer (PROXIMA 1022/1/1, ALLIANCE instruments, France). Soil ammonium ($NH_4^+$-N, AN) content was extracted using 2 M KCl on an orbital shaker for 1 h under ambient temperature and then the suspension was filtered. The extracts were analysed by a continuous flow analytical system (SKALAR San++, SKALAR Co., Netherlands). Soil available phosphorus (AP) was determined by the ammonium chloride-hydrochloric acid extraction method (*Bao, 2000*), and acid phosphatase activity (APA) was determined by the phosphoric acid bisacid colourimetric method (*Guan, 1986*). Soil moisture (SM) was represented as the ratio of dry soil weight to soil water weight after fresh soil samples were dried at 105 °C for 24 h. Soil pH was detected by the potentiometric method, where 10 g air-dried soil was put into a 25 ml beaker and added with distilled water and kept for 30 min, after which the suspension's pH value was measured with a corrected pH meter. Soil particle size distributions were determined by wet sieving (*Chaudhari, Singh & Kundu, 2008*). To investigate the effects of local topography on soil properties, we binned each data point into a topography type based on elevations, slopes and convexities

by fuzzy *C*-mean clustering cluster analysis (*Xu et al., 2015*). Four topography types were identified: a flat area, a relative steep area, a steep area and a mountaintop area. We then performed one-way ANOVAs to test the differences in the soil properties among the topographic groups. The soil property was regarded as significantly different if *P*-value was ≤0.05. All results were shown as mean ± SE. Path analysis provides a way to examine the multiple relationships between SOC stability and environmental factors. Thus, a path analytic framework was applied to examine the influencing factors on SOC stability across the different topographies. As soil nutrients and other soil properties did not significantly differ among the four topographies. The flat and the top area were classified into the relative flat (RF) areas and the rest two types into the relative steep (RS) areas according to *Wang et al. (2018)*. The RF areas had more favorable nutrients and environment than the RS areas and could be applied to validate the hypothesis we proposed previously.

All analyses were conducted using SPSS Version 20 and figures were generated by Origin (version 92 E).

# RESULTS

## Soil properties in the different topography types

Soil properties varied among the different topographies. To be specific, SOC increased from flat areas to mountaintop areas, with 29.66 g/kg in flat areas and 41.97 g/kg in mountaintop areas. Soil TN tended to increase in mountaintop areas, with 1.47 g/kg in flat areas and 1.78 g/kg in mountaintop areas. However, soil C:N ratios were similar across the different topographies. The AN, TP and SM were also the highest in the mountaintop areas, whereas the contents of pH was highest in the flat areas. Soil particle size did not change with topography except slit content was high in the flat area (Fig. 2 and Table 1).

## SOC components in the different topography types

Alkyl carbon, on average 41.67% of the SOC, was the largest component of the SOC chemical fractions in the four different topography types (Fig. 3). Aromatic carbon, which accounted for only 2.61%, was the lowest SOC component. O-alkyl carbon and carboxyl carbon took up 33.08% and 22.80% of the SOC, respectively. As illustrated in Fig. 4, the overall distributions of the fractions of the SOC were similar across the different topography types. However, the proportion of alkyl carbon and carboxyl carbon was significantly altered by topography. The alkyl carbon content increased from the flat to mountaintop areas, with 38.87% in flat areas and 43.34% in mountaintop areas, while the carboxyl carbon content decreased from 24.78% in flat areas to 21.24% in mountaintop areas. O-alkyl carbon and aromatic carbon content did not change significantly with topography.

## SOC stability indices in the different topography types

In all four topographies, the DI was higher than 1 and the II was higher than 0.7. The DI was not significantly different among the topographies, while II was significantly different in different topographies with the lowest II value in the flat areas (0.72) and the highest value in the steep areas (0.91) (Fig. 5).
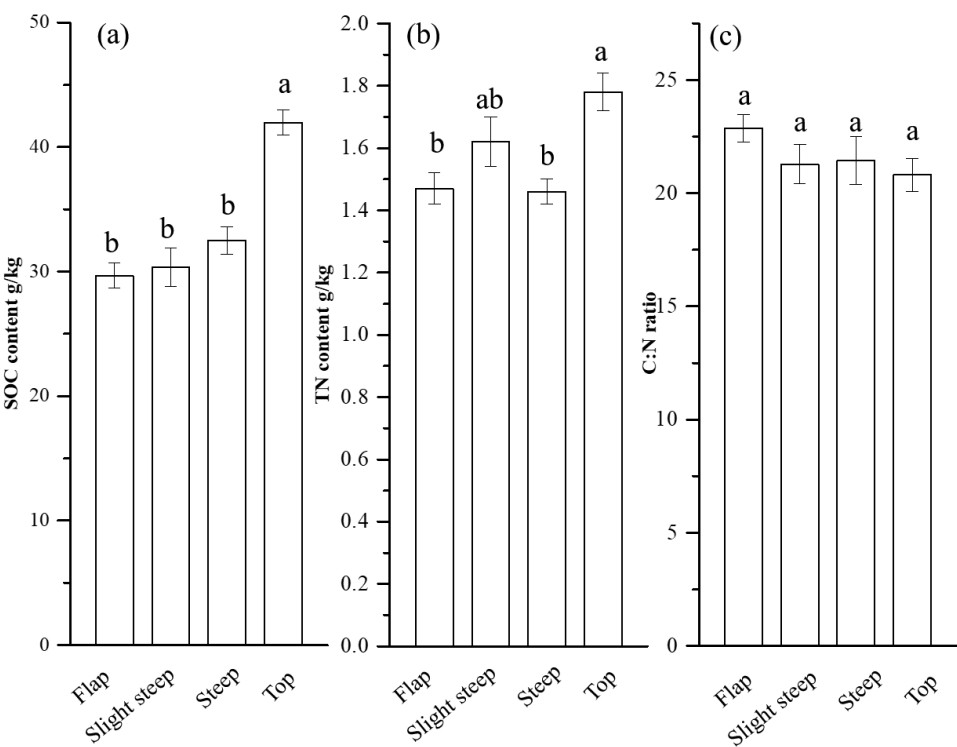

**Figure 2** **The changes of soil SOC (A), TN (B) and C:N ratio (C) among four topographies.** The different lowercase letters indicate significant differences among topographies ($P < 0.05$).

**Table 1** **Soil properties in different topographic areas ($n = 60$).**

| Variables | Flat ($n = 12$) | Slight steep ($n = 16$) | Steep ($n = 13$) | Top ($n = 19$) | Average ($n = 60$) |
|---|---|---|---|---|---|
| AN (mg/kg) | 30.29 ± 6.04bc | 42.16 ± 4.32ab | 29.21 ± 5.69c | 53.03 ± 3.41a | 40.40 ± 2.62 |
| TP (mg/kg) | 74.12 ± 5.64b | 82.50 ± 5.62ab | 77.66 ± 5.07b | 90.91 ± 3.68a | 82.40 ± 2.54 |
| AP (mg/kg) | 2.71 ± 0.18a | 3.03 ± 0.10a | 2.92 ± 0.08a | 2.30 ± 0.11a | 2.93 ± 0.06 |
| C:P ratio | 468.65 ± 34.09a | 433.61 ± 25.56a | 442.74 ± 43.57a | 407.08 ± 24.19a | 434.20 ± 15.36 |
| N:P ratio | 20.56 ± 1.42a | 20.26 ± 0.66a | 20.39 ± 1.46a | 19.68 ± 0.96a | 20.16 ± 0.54 |
| SM | 20.88 ± 1.66cb | 21.20 ± 1.52b | 16.86 ± 0.83dc | 25.28 ± 1.22a | 21.49 ± 0.77 |
| pH | 4.42 ± 0.05a | 4.32 ± 0.05b | 4.31 ± 0.04ab | 4.15 ± 0.02c | 4.28 ± 0.02 |
| Sand (%) | 45.90 ± 4.81a | 46.13 ± 3.28a | 47.23 ± 2.72a | 40.08 ± 2.20a | 44.10 ± 1.61 |
| Silt (%) | 27.42 ± 3.57b | 33.60 ± 2.22a | 32.61 ± 2.85a | 40.24 ± 2.51a | 34.53 ± 1.51 |
| Clay (%) | 26.68 ± 6.58a | 20.27 ± 4.14a | 20.16 ± 4.13a | 19.68 ± 3.18a | 21.37 ± 2.17 |

**Notes.**
Abbreviation: AN, was ammoniacal nitrogen; TP, was total phosphorus; AP, was available phosphorus; C:P ratio, was calculated as SOC/ TP; N:P ratio, was calculated as TN/TP; SM, was soil moisture; n, was sample size.
All values were expressed as mean values ± SE. The different lowercases mean significant differences among topographies ($P < 0.05$).

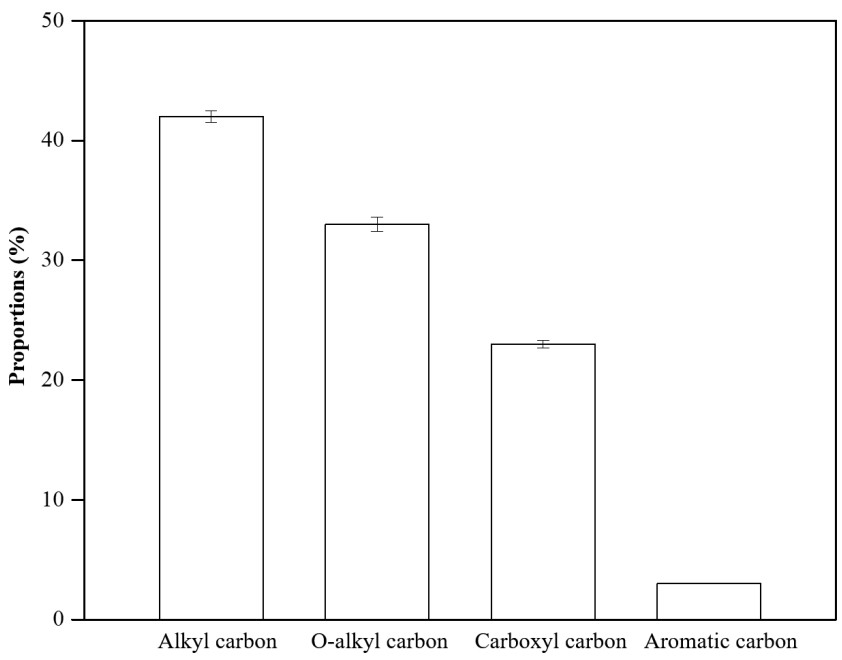

**Figure 3** **The distributions of SOC components in this tropical rainforest.** Errors bars represent the standard error (SE).

## Influencing factors on SOC stability in the different topography types

Figure 6 shows that the influencing factors and mechanisms of SOC stability were different in the different topography types. In the RF areas, SOC, AN and SM together explained 52% of the variability in the SOC stability. SOC and AN directly positively affected SOC stability; high SOC and AN strengthened SOC stability. SM directly negatively influenced SOC stability, however, the direct negative effect of SM was partly offset by the indirect positive effect of SM on AN and SOC, thus, SM had little overall effects on SOC stability. Besides, SOC were positively correlated with soil silt content in this area (Fig. 7). However, soil particle size was not adopted by the model.

In the RF areas, SM and AN together explained 26% variability of SOC stability. In the RS areas, SM and AN also affected SOC stability by 42%. The effect was the same as for RF areas, though the effect of SM was larger than that of AN on SOC stability. Increasing SM decreased SOC stability, though this negative effect was partly offset bySM's indirect effect on AN. AN directly positively affected SOC stability.

Soil AN and SM were influencing factors on SOC stability in the RF areas and the RS areas, however, the extent of their effect differed. Specifically, SM was a negligible limiting factor in the RF areas, whereas SM was the most important limiting factor in the RS areas. The path analyses model had an R square of 0.53 in the RF areas, while this was 0.42 in the RS areas (Table S1).

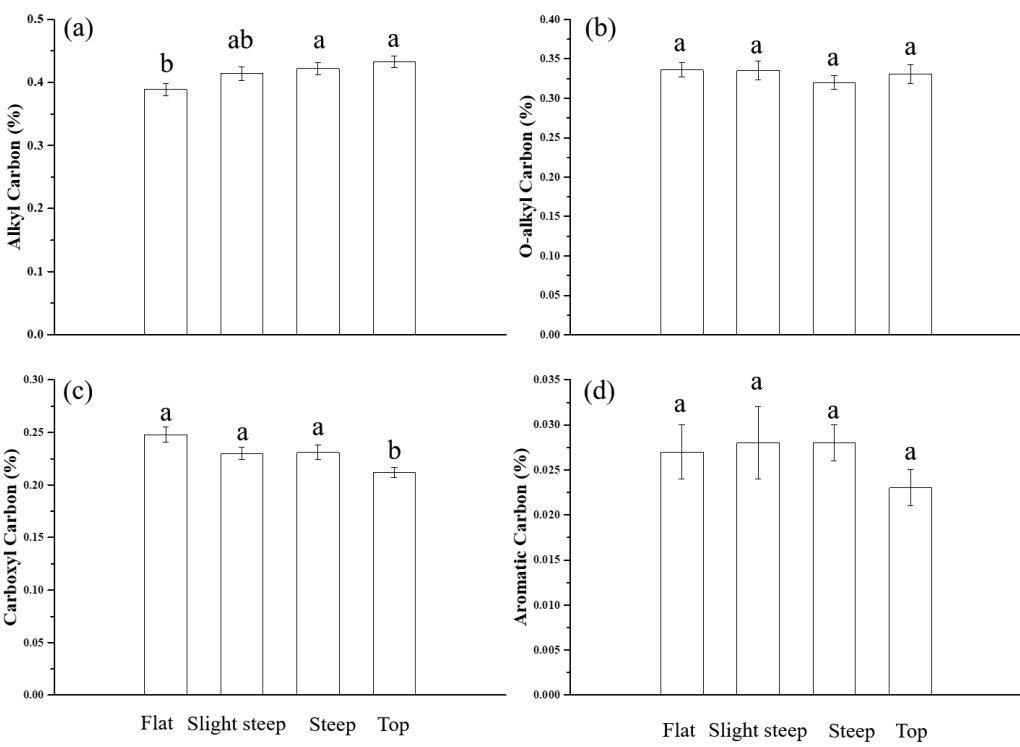

**Figure 4** **The comparisons of SOC fractions in the four topography types.** (A) Alkyl carbon, (B) O-alkyl carbon, (C) carboxyl carbon, and (D) aromatic carbon. Error bars represent the standard error. The different lowercases mean significant differences among topographies ($P < 0.05$).

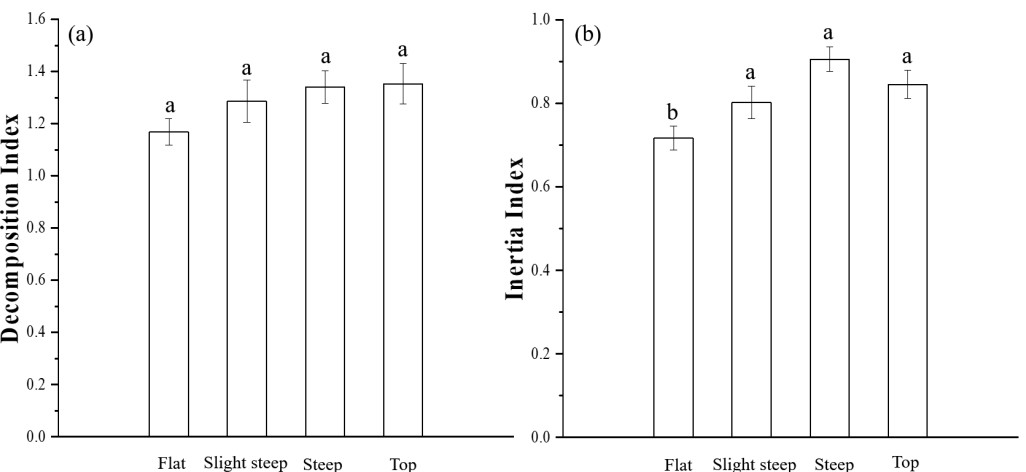

**Figure 5** **Difference of decomposition index (DI) and inertia index (II) among topographies.** (A) decomposition index, (B) inertia index. Error bars represent the standard error. The same letter means variables in the different topography types are not significantly different.

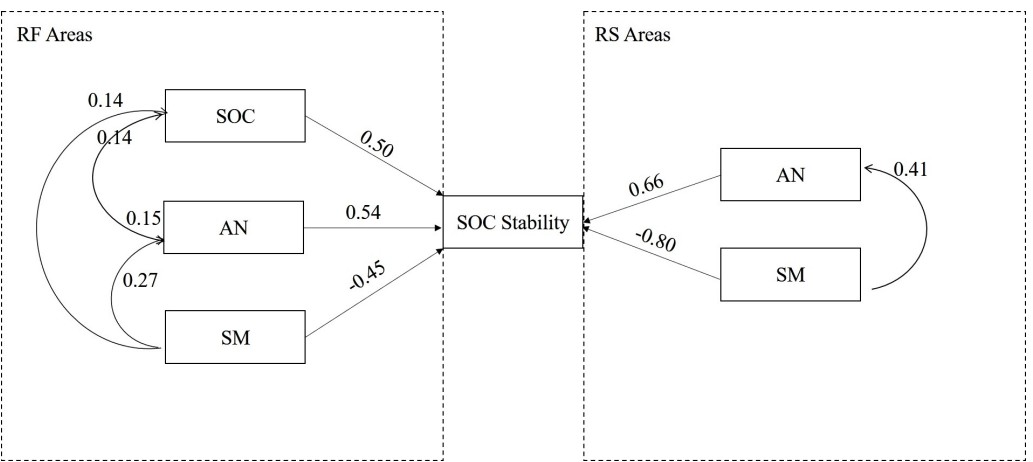

**Figure 6 Influencing factors and mechanisms of SOC stability in two topographies.** Values aside the straight line are direct path coefficient, and values aside the curve are indirect path coefficient. The direction of arrow indicates the effect direction.

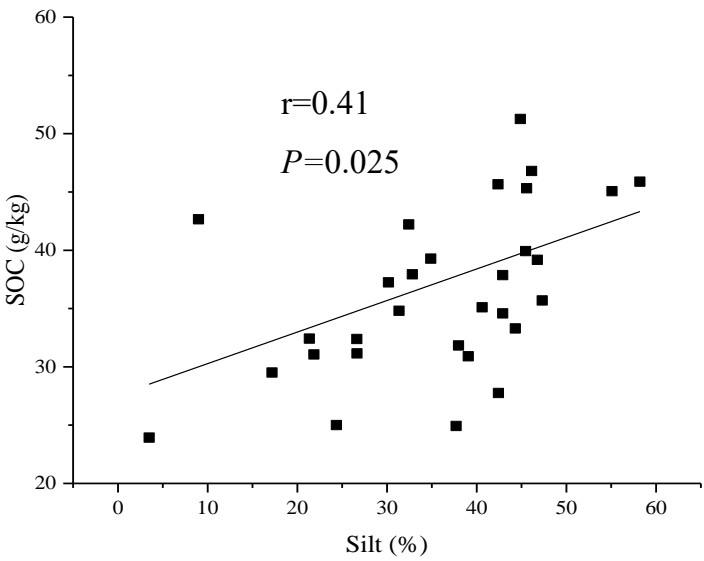

**Figure 7 The simple liner relationship between soil texture and SOC content in the relative flat areas.**

## DISCUSSION

The chemical fractions of the SOC in the different topographic areas were characterized by alkyl carbon > O-alkyl carbon > carboxyl carbon > aromatic carbon. The ranking of the SOC fractions is similar to those of subtropical evergreen board-leaved forest and Masson pine in China (*Shang et al., 2013*), and subtropical natural forest and hoop pine plantation in Australia (*Chen, Xu & Mathers, 2004*) with 13CNMR technology. Alkyl carbon was the largest component of the SOC fractions, indicating that the SOC comprised a higher percentage of passive carbon in SOC. The DI (alkyl carbon/O-alkyl carbon)

for the 60 ha was 1.29, which is higher than moist semi-deciduous tropical forest soils in Ghana (1.25) (*Elberling, Breuning-Madsen & Knicker, 2013*), Costa Rican old live oak forest (0.87) and old-growth dry tropical forests (0.70) (*Lorenz, Lal & Jiménez, 2010*), plantations in the subtropics of China (Pinus massoniana, Castanopsis hystrix, Michelia macclurei, Mytilaria laosensis, all < 0.8) (*Wang, 2010*), evergreen broad-leaved forest (0.76), Cunninghamia lanceolata forest (0.89), Cryptomeria fortune forest (0.78), and coniferous and broad-leaved mixed forest (0.53) in the subtropics of China (*Zhang et al., 2015*). The DI of all the different topography types was all higher than 1, indicating a rather stable SOC in the tropical montane rainforest of Jiangfengling, Hainan Island (*González-Pérez et al., 2008*). Studies of changes in SOC and its fractions of forest along a climatic gradient in China found sites with higher mean annual temperature would have greater alkyl C proportion and higher DI value as active SOC decomposition processes accelerated (*Sun et al., 2019*). Path analysis showed that the SOC contents only affected SOC stability in RF areas, while AN and SM affected SOC stability in both topographies. Studies have found that interactions between SOC and mineral surfaces can stabilise SOC, whereby stable organic-mineral bonds were formed through anion and inner-sphere ligand-exchange reaction (*Wiesmeier et al., 2019*). Tropical forest with highly weathered soils contains high concentrations of iron (*Sanchez, 1976*) that could protect C from microbes and enhance SOC stability. Studies found $Fe^{3+}$ was positively correlated with the SOC content (*Wang et al., 2019*). The SOC affected SOC stability in the RF areas might be due to the fact that SOC content in the RF areas was significantly higher than that in the RS areas (37.21 > 31.32 g/kg, $p < 0.01$).

SOC stability depends not only on its chemical characteristics, but also on the soil microenvironment (*Yang et al., 2020*). Soil biotic community and the soil microenvironment are key factors that affecting the SOC stability (*Schmidt et al., 2011*; *Yang et al., 2018*). Soil N, especially in the bioavailable forms, such as ammonium is an essential component of all living organisms. Soil N dynamics are mainly driven by microbes (*Levy-Booth, Prescott & Grayston, 2014*) and microbes mainly assimilate of AN (Cheng et al., 2015), however, some studies have found that nitrogen addition negatively affects soil microbial growth, composition and function (*Chen et al., 2018*). The soil AN content in the study area was rather high at about 40.4 mg/kg while the value in Amazon pristine forest was 5.7 mg/kg (*Hamaoui et al., 2016*) and the value in a secondary tropical forest in Southern China was 2.12 mg/kg (*Wang et al., 2014*). Therefore, microorganism activities and functions might be suppressed by the high AN content in this area, resulting in a slow decomposition rates and consequently high SOC stability in this tropical forest.

Soil moisture is a key factor influencing SOC mineralization in terrestrial ecosystems (*Liu, Zhang & Wan, 2009*; *Moyano, Manzoni & Chenu, 2013*). Soil moisture can affect SOC decomposition directly or indirectly. Topography influences soils by the transport of fine soil particle towards the base of slopes and lower slope positions in what are called depositional areas, which tend to have high organic materials and water-holding capacity (*Chapin III, Matson & Vitousek, 2012*). The above process results in the deposition of more active carbon than passive carbon, explaining the low DI and II in the RF areas. This explains why SM was a negative influencing factor of SOC stability in the RS and RF areas (Fig. 6).

Moreover, the SOC in the flap areas was the most unstable among the four topography types (Fig. 4). In terms of indirect effects, decomposition of soil organic matter depends on factors such as soil mineralogy, redox potential and electron acceptor availability, which are controlled by soil water regimes (*Ro, Ji & Lee, 2018*). Studies have found that an increase in soil water content benefits dissolved organic carbon dissolution and nutrients transfer, which stimulates the activity of microorganisms involved in organic carbon decomposition (*Goebel et al., 2007*). Higher SM results in higher rates of substate supply to the microbes and, thereby, higher microbial growth. In addition, soil moisture availability can affect the vegetation distribution and structure (*D'Odorico et al., 2007*; *Ruiz-Sinoga et al., 2011*), affecting the amount and quality of litter, which in turns influences the soil organic matter decomposition rate (*Rodríguez-Iturbe & Porporato, 2007*). On the other hand, AN is water-soluble nitrogen which is vulnerable to the changes of soil water. Hence, SM could negatively affect SOC stability by positively influencing the soil microbes and AN content.

# CONCLUSIONS

Our analyses of the effects of small-scale topography on SOC stability in a tropical mountain rain forest showed that topographic heterogeneity altered SOC stability by changing soil nutrients and soil moisture. Our findings indicated that SOC is relatively stable in the tropical montane rainforest topsoil and AN and SM were the main influencing factors of SOC stability both in RF and RS areas. Furthermore, SOC and AN positively promote SOC stability while SM was a limiting factor on SOC stability in this study areas. Accordingly, soil nutrients were the key influencing factors in RF areas but not in RS areas. Our hypothesis that nutrients and soil environment were the key influencing factor in studied areas were partly confirmed. The path analysis models explained less than 50% of the SOC stability in the RS areas, thus the mechnism remains to be further explored.

# ACKNOWLEDGEMENTS

We are grateful to Jianfengling National Key Field Research Station for Tropical Forest Ecosystem for providing their assistance in field work.

## Funding

This work was supported by the Natural Science Foundation of Hainan province [No.2019RC012 and 418MS019]; the National Key R&D Program of China [NO. 2018YFD0201105]; the National Natural Science Foundation of China [No. 41663010 and 41201061]. The funders had no role in study design, data collection and analysis, decision to publish, or preparation of the manuscript.

## Grant Disclosures

The following grant information was disclosed by the authors:
The Natural Science Foundation of Hainan province: 418MS019, 2019RC012.

The National Key R&D Program of China: 2018YFD0201105.
The National Natural Science Foundation of China: 41663010, 41201061.

## Competing Interests

The authors declare there are no competing interests.

## Author Contributions

- Yamin Jiang performed the experiments, analyzed the data, prepared figures and/or tables, authored or reviewed drafts of the paper, and approved the final draft.
- Huai Yang and Wenjie Liu conceived and designed the experiments, authored or reviewed drafts of the paper, and approved the final draft.
- Qiu Yang performed the experiments, prepared figures and/or tables, and approved the final draft.
- Zhaolei Li analyzed the data, authored or reviewed drafts of the paper, and approved the final draft.
- Wei Mao and Xu Wang performed the experiments, authored or reviewed drafts of the paper, and approved the final draft.
- Zhenghong Tan conceived and designed the experiments, authored or reviewed drafts of the paper, and approved the final draft.

## Data Availability

The raw measurements are available in the Supplemental File.

## Supplemental Information

Supplemental information for this article can be found online at http://dx.doi.org/10.7717/peerj.12057#supplemental-information.

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
