# Peer review of "The stability of soil organic carbon across topographies in a tropical rainforest"

_PeerJ, doi:10.7717/peerj.12057_

## Round 0.1 · original submission · Major Revisions

Dear Dr. Jiang and co-authors,

I just received reviews of your manuscript. Although both reviewers consider the study very interesting and significant, some issues need to be considered before acceptance.

While reviewer #1 highlights the need to add the soil texture data to better analyze the influencing factors and finds numerous errors or inconsistencies that must be corrected, reviewer#2 emphasizes the lack of data or results analysis in each part of the manuscript. Please, consider all comments and suggestions provided by both reviewers during the revision of your manuscript.

A comprehensive revision of the English of the manuscript is necessary before submitting the new version.

Don't forget to include a letter response along with the revised version of the manuscript. In this letter, you must respond point by point to each question.

Best regards,

Xiaoming Kang

Reviewer 1 ·

Basic reporting

English writing should be improved.
physical protection of SOC such as soil texture should be included.

Experimental design

this manuscript is within the aims and scope of the journal.
the research question is meaningful for understanding SOC stability in tropical rainforest.
the research method is suitable for the question and the description of method is sufficient.

Validity of the findings

Conclusions are well stated.

Additional comments

Comments:
This manuscript explored the underlying mechanism of topography effects on chemical fraction of SOC and SOC stability in Jianfengling, Hainan island. The authors found that the chemical fractions proportion of SOC showed the trend of alkyl carbon > O-alkyl carbon > carboxyl carbon > aromatic carbon. The decomposition index were greater than 1 in different topography types. Moreover, the influencing factors of SOC stability were SOC content, soil moisture, and ammonia nitrogen in relative flat area (including flat and slight steep topography) and soil moisture, and ammonia nitrogen in relative steep area (including steep and top topography). Overall, these findings are helpful to understand the mechanism underlying the effects of topography on SOC chemical fractions and stability in tropical rainforest.

The experimental method is suitable for this study and could provide support for the conclusion. The topic is meaningful and provide potential mechanism involved in the effects of topography on SOC chemical fractions and stability. However, the major concern of this manuscript is that the authors only consider the chemical factors including SOC content, ammonia nitrogen and neglect physical protection of SOC including soil texture (sand, loam, clay proportion). I suggest that the authors should add the soil texture data to re-analyze the influencing factors. In addition, the authors should substantially improve English writing and I therefore recommend it for publication after major revision.

Specific comments:
1. Abstract section. The authors should add results of differences of four different topographies not only in RF and RS.
2. Line 17, “the chemical fractions of SOC showed …”the meaning is not clear.
3. Line 20-23. The influencing factors of SOC ….in the relative steep areas. Please change this sentence into short sentences.
4. Line 24, Inertia index also higher in Rs than that in RF. This sentence is wrong in grammar.
5. Line 41-42. Add reference behind this sentence.
6. Line 57. “Studies found topography influences nutrients availability…” problems in grammar.
7. In introduction section, the authors should add the effects of physical protection on SOC stability.
8. Line 92. Fe3+ and Mn2+
9. Line 134. Is this classification into two types eligible? Please verify it from references.
10. Line 156. Should describe the differences of o-alkyl carbon and aromatic carbon content between topography types.
11. Line 169. Should add “AN and SM explained **% variability of SOC stability in RF areas”
12. Line 180. “ are characterized by the trend with alkyl carbon…”
13. Line 199-200. The SOC only affected SOC stability in RF areas might be due to the SOC content in RF areas was significantly higher than that in the RS areas. Is this description contrary to Fig. 2? Fig.2 showed that steep and top had the highest SOC content.
14. Line 219. “Stubborn” here is not suitable. Change into another word.
15. Line 224-226. Studies found increase soil water content benefits dissolved organic carbon dissolution….. in organic carbon decomposition. This sentence has grammar problem.
16. Line 241. “We hypothesized nutrients be the key influencing factor in study areas was not fully confirmed.” Please paraphrase this sentence.

Annotated reviews are not available for download in order to protect the identity of reviewers who chose to remain anonymous.

Reviewer 2 ·

Basic reporting

Stable mechanisms of tropical rainforest soil organic carbon were still unclear. In addition, this study is interesting and significant, I recommend reconsideration of this manuscript following minor revision and modification. Furthermore, this manuscript needs a linguistic revision before it is resubmitted and it is suggested that the authors seek some professional help in doing this.

Experimental design

This study analyzed the chemical fractions of SOC by novel 13C CPMAS/NMR technology. Driving factors of SOC stable process were revealed under different topographies. Experimental design and statistical analysis was correct.

Validity of the findings

Abstract, some abbreviates were not necessary, and it was redundant. “soil was” changed to soils were
Introduction, the second paragraph lacked specific data regarding soil organic carbon fraction and its stability under different topographies. Moreover, this section should add its potential driving factors.
Discussion section, please adding some relative results by 13C CPMAS/NMR technology.

---

## Round 0.2 · Minor Revisions

Dear Dr. Jiang and co-authors,

I just received reviews of your manuscript. Some small problems need to be solved before the acceptance.

Don't forget to include a letter response along with the revised version of the manuscript. In this letter you must respond point by point to each question.

Best regards,

Xiaoming Kang

Reviewer 1 ·

Basic reporting

literature references should be made carefully.

Experimental design

no comment

Validity of the findings

no comment

Additional comments

Comments:
The English writing and the text structure have been improved in the revised manuscript. I have no further comments on this manuscript, but modifications of references citation formation should be made before finally accepted.
The authors should especially pay attention to the references with 3 authors.
Line 53. ChenXu & Mathers 2004
Line 114. ChaudhariSingh & Kundu 2008
Lin3 199. Chapin IIIMatson & Vitousek 2012

Reviewer 2 ·

Basic reporting

Thanks to the careful and powerful revises of authors. I insist that the manuscript could be accepted at this form.

Experimental design

satisfied.

Validity of the findings

satisfied.

---

## Round 0.3 · accepted · Accept

Dear authors,

I am pleased to inform you that, following the revision made based on the reviewer’s comments, your manuscript is now acceptable for publication in PeerJ.

Best regards

Xiaoming Kang